# Doxepin Exacerbates Renal Damage, Glucose Intolerance, Nonalcoholic Fatty Liver Disease, and Urinary Chromium Loss in Obese Mice

**DOI:** 10.3390/ph14030267

**Published:** 2021-03-16

**Authors:** Geng-Ruei Chang, Po-Hsun Hou, Wei-Cheng Yang, Chao-Min Wang, Pei-Shan Fan, Huei-Jyuan Liao, To-Pang Chen

**Affiliations:** 1Department of Veterinary Medicine, National Chiayi University, 580 Xinmin Road, Chiayi 60054, Taiwan; leowang@mail.ncyu.edu.tw (C.-M.W.); babybelle349@gmail.com (P.-S.F.); pipi324615@gmail.com (H.-J.L.); 2Department of Psychiatry, Taichung Veterans General Hospital, 1650 Taiwan Boulevard (Section 4), Taichung 40705, Taiwan; peterhopo2@yahoo.com.tw; 3Faculty of Medicine, National Yang-Ming University, 155 Linong Street (Section 2), Taipei 11221, Taiwan; 4School of Veterinary Medicine, National Taiwan University, 1 Roosevelt Road (Section 4), Taipei 10617, Taiwan; yangweicheng@ntu.edu.tw; 5Division of Endocrinology and Metabolism, Show Chwan Memorial Hospital, 542 Chung-Shan Road (Section 1), Changhua 50008, Taiwan

**Keywords:** doxepin, obesity, chromium, fatty liver disease, renal impairment, glucose intolerance

## Abstract

Doxepin is commonly prescribed for depression and anxiety treatment. Doxepin-related disruptions to metabolism and renal/hepatic adverse effects remain unclear; thus, the underlying mechanism of action warrants further research. Here, we investigated how doxepin affects lipid change, glucose homeostasis, chromium (Cr) distribution, renal impairment, liver damage, and fatty liver scores in C57BL6/J mice subjected to a high-fat diet and 5 mg/kg/day doxepin treatment for eight weeks. We noted that the treated mice had higher body, kidney, liver, retroperitoneal, and epididymal white adipose tissue weights; serum and liver triglyceride, alanine aminotransferase, aspartate aminotransferase, blood urea nitrogen, and creatinine levels; daily food efficiency; and liver lipid regulation marker expression. They also demonstrated exacerbated insulin resistance and glucose intolerance with lower Akt phosphorylation, GLUT4 expression, and renal damage as well as higher reactive oxygen species and interleukin 1 and lower catalase, superoxide dismutase, and glutathione peroxidase levels. The treated mice had a net-negative Cr balance due to increased urinary excretion, leading to Cr mobilization, delaying hyperglycemia recovery. Furthermore, they had considerably increased fatty liver scores, paralleling increases in adiponectin, FASN, PNPLA3, *FABP4* mRNA, and *SREBP1* mRNA levels. In conclusion, doxepin administration potentially worsens renal injury, nonalcoholic fatty liver disease, and diabetes.

## 1. Introduction

Doxepin is a dibenzoxepine-derived sedating tricyclic antidepressant approved for treating various psychiatric conditions, including insomnia, disrupted sleep patterns, anxiety, bipolar and attention-deficit hyperactivity disorders, autism, depression, and schizophrenia [1,2]. In 1969, doxepin was indicated for the first time to treat major depressive and anxiety disorders in the United States, where it is still accounts for >2 million annual prescriptions [2]. Doxepin inhibits norepinephrine/serotonin reuptake within synaptic clefts in the central nervous system and, thus, increases these neurotransmitters’ levels in the brain [1]. In animal behavioral models, doxepin has shown effective results by demonstrating antidepressive activities such as suppression of chronic stress (electric shock)–induced depressive behavior [3]. Moreover, in a preclinical study, a combination of doxepin with various other pharmacological agents demonstrated synergistic effects in reducing the marble-burying behavior of mice [4]. Furthermore, doxepin may act as a nonselective serotonin receptor/reuptake inhibitor for the serotonin transporter [5]. Another beneficial effect of a strong serotonin antagonist such as doxepin is that it may counteract the most serious, even lethal, side effects of some antidepressants (e.g., serotonin syndrome, which has a ≤5.9% mortality rate), particularly when administered alongside serotonin reuptake inhibitors to treat mental disorders [6].

Patients under tricyclic sedating antidepressant treatment, however, may exhibit memory impairment, dry mouth, drowsiness, fine tremor, micturition difficulty, constipation, postural hypotension, dyslipidemia, and weight gain [7,8,9]. Many commonly used psychotropic medications, particularly antipsychotics, and antidepressants, have recently been independently associated with endocrine and metabolic factors such as insulin resistance (IR), obesity, and dyslipidemia [8,10]. Triglycerides possibly represent a major factor in the pathogenesis of IR, the most crucial type 2 diabetes mellitus (T2DM) predictor [11,12]. Moreover, nonalcoholic fatty liver disease (NAFLD) may lead to triglyceride overproduction in the liver due to hyperinsulinemia and glucose intolerance (both commonly observed along with IR) [13,14]. Patients with schizophrenia are highly likely to develop liver disease, including NAFLD, chronic liver disease, and alcohol-related cirrhosis [15,16]. Moreover, this exhibition increases these patients’ susceptibility to sharp post–antipsychotic treatment increases in triglyceride levels, which result in liver injury. Furthermore, among patients with incident schizophrenia, the increased chronic kidney disease (CKD) risk potentially has an association with metabolic syndrome due to antipsychotic treatment [8,17]. This increase may be mild or moderate in individuals receiving antipsychotic treatment. However, whether the use of typical/atypical antipsychotics is associated with the risk of kidney damage in obese patients with psychotic disorders remains unclear [18,19].

The side effects—including tiredness, headache, palpitation, perspiration, constipation, as well as sleep- and micturition-related problems [20,21]—of tricyclic antidepressants may reduce patient compliance and thus increase health care costs and exacerbate the physical and financial burdens on patients themselves and their caregivers [8,9]. However, in comparison with different antipsychotics (e.g., clozapine, risperidone, olanzapine, and haloperidol), doxepin treatment is less likely to alter the patient’s physical parameters (e.g., body weight and lipid profile) in the long term [22,23,24]. Moreover, in mice, long-term doxepin administration was noted to reduce body and white adipose tissue (WAT) weights—an observation inconsistent with the findings of most human studies [25]. A relevant study excluded a case report on a T2DM patient with hypoglycemia because the patient was taking doxepin concurrently with sulfonylurea [26]. Moreover, in another case report, the patient was noted to have experienced hypoglycemia within 11 days of starting doxepin treatment [27]. Furthermore, doxepin-treated animals have been noted to have increased insulin sensitivity (IS); thus, doxepin may regulate glucose metabolism through Akt activation enhancement and gluconeogenesis suppression [28,29]. However, in patients using antipsychotics, doxepin can lead to hyperglycemia and exacerbate preexisting diabetes, hypertriglyceridemia, and hypercholesterolemia [30,31,32], but it can be an effective treatment option for postprandial hypoglycemia [33]. Long-term doxepin use can also induce NAFLD-associated body weight gain [34] and liver injury [35]. Antipsychotics, including doxepin, aripiprazole, fluphenazine, risperidone, nitrazepam, olanzapine, and ziprasidone, may cause medication-associated nephrotoxicity; thus, concerns regarding whether these drugs may increase the risk of kidney injury (e.g., interstitial nephritis, glomerulonephritis, and CKD) have been raised [8,36].

Taken together, the observed differential effects of doxepin in terms of body weight and hyperglycemia alterations may result from differences according to the presence and severity of metabolic syndrome. Therefore, in the present study, animals with high-fat diet (HFD)–induced obesity received doxepin treatment; this model was designed to mimic doxepin’s effects in individuals with obesity who also have psychotic disorders. Moreover, most studies thus far have explored doxepin’s adverse effects on metabolism by focusing on alterations in how lipids and glucose are metabolized; consequently, the influence of doxepin on glucose homeostasis–related changes in fatty liver, kidney damage, and levels of trace elements, particularly chromium (Cr), remains unclear. Moreover, supplementation with Cr—which is involved in normal carbohydrate, lipid, and protein metabolism—has beneficial effects in people with glucose intolerance, DM, obesity, or nephropathy [37]. Thus, we sought to determine the metabolic effects doxepin has on the hepatic, pancreatic, and renal function of obese animals. Our current results are likely to provide further key insights into how the long-term use of doxepin as a psychotropic agent affects metabolism and causes fatty liver and kidney damage and reveal whether long-term doxepin administration exacerbates liver and kidney injury as well as metabolic abnormalities.

## 2. Results

### 2.1. Effects on Morphometric Parameters and Food Intake and Efficiency

In our preliminary research on standard diet (SD)–fed mice, the anti-obesity effects of doxepin treatment were nonsignificant (Appendix A). By contrast, among the mice on an HFD, those receiving eight-week doxepin treatment had higher metabolic morphometric parameters and serum leptin levels than did those receiving a saline control (Figure 1). In general, doxepin treatment led to significant weight gain; compared with the control mice, the doxepin-treated mice demonstrated 1.2-, 2.2-, and 2.3-fold higher body weight (Figure 1a), body weight change (Figure 1b), and body weight gain (Figure 1c), respectively. Moreover, with regard to food intake, the treated mice demonstrated significant increases in weekly food intake (1.4-fold; Figure 1d) and daily food efficiency (2.1-fold; Figure 1e) compared with the controls. The mice receiving doxepin treatment also had three-fold higher serum levels of leptin (Figure 1f)—which is involved in food intake regulation [38].

### 2.2. Effects on Liver, Kidney, and WAT Weights

Next, we evaluated the potential relationship of the observed weight differences with alterations in body composition or adiposity. Following 56 treatment days, the treated and control mice demonstrated significantly different liver, kidney, epididymal WAT (EWAT), and retroperitoneal WAT (RWAT) weights (Figure 2): the treated mice had 26%, 12%, 63%, and 24% higher liver, kidney, RWAT, and EWAT weights, respectively. Moreover, as percentages of the total body weight (TBW; i.e., absolute weight), the mice in the treatment group had liver, kidney, RWAT, and EWAT weights that were, respectively, 16%, 14%, 43%, and 15% higher than those among mice in the control group. However, the kidneys and spleen weights (general and absolute weights) did not differ between the treated and control mice.

### 2.3. Effects on Liver Fat Accumulation and Large Adipocyte Proportion

We subsequently performed a morphometric analysis using hematoxylin and eosin (H&E)-stained tissues of the control and treated mice. The results demonstrated that the treated mice had significant increases in the sizes of RWAT and EWAT adipocytes and the levels of liver fat (Figure 3a). Thus, doxepin possibly prevents liver fat accumulation, consequently increasing WAT hypertrophy. Further comparative analysis indicated that the mice in the treatment group had nearly 1.5-fold higher fatty liver scores (Figure 3b) than those of the controls. Moreover, the mean RWAT (Figure 3c) and EWAT (Figure 3d) adipocyte sizes were (significantly) 15% and 18% larger in the treated mice, respectively, thus corroborating the observation of relatively high WAT weights in the treated mice. In other words, fewer adipocytes with diameters of 0–5 and 50–100 μm, but more with diameters of 100–150 and >150 μm were observed in the treatment group (Table 1). Therefore, doxepin accelerated the considerable growth in EWAT and RWAT adipocytes induced by an HFD.

### 2.4. Effects on Serum ALT and AST Levels and Fatty Acid–Binding Protein 4 and Sterol Regulatory Element–Binding Protein 1 mRNA Levels

We next evaluated serum ALT and AST levels (both of which are liver function markers) in the treated mice and noted 35% (Figure 4a) and 30% (Figure 4b) increases compared with the control mice.

Fatty acid–binding protein 4 (FABP4) and sterol regulatory element–binding protein 1 (SREBP1) have been implicated to have regulatory roles in numerous metabolic pathways, including lipid accumulation in the liver, T2DM, and atherosclerosis [10,39]. Consequently, these proteins have predominant functions in promoting fatty liver disease in both humans and rodents [40]. Here, *FABP4* and *SREBP1* mRNA levels in the liver (Figure 4c,d, respectively) were 1.8- and 2.5-fold higher in the treatment group than in the control group, respectively.

Therefore, we believe that doxepin contributed to fatty liver disease development through the activation of liver steatosis genes (*FABP4* and *SREBP1*), which then led to significant increases in ALT and AST levels.

### 2.5. Effects on Serum and Liver Triglyceride Levels and Fatty Acid Synthase, Adiponectin, and Patatin-Like Phospholipid Domain Containing Protein 3 Expression

Adiponectin not only mediates the stimulation of energy expenditure but also regulates the expression of glucose and lipid metabolism–related liver genes [40]; fatty acid synthase (FASN) mainly regulates triglyceride synthesis and lipid homeostasis [41]; and finally, patatin-like phospholipid domain containing protein 3 (PNPLA3) regulates lipogenesis in not only obesity and NAFLD but also cardiovascular disease [10]. Therefore, we obtained mouse liver extracts and performed Western blot analysis for adiponectin, FASN, and PNPLA3 expression (Figure 5c). Notably, in comparison with the controls, the mice in the treatment group had 43% lower liver adiponectin expression levels (Figure 5d) but 63% and 75% higher liver FASN (Figure 5e) and PNPLA3 (Figure 5f) expression levels, respectively.

Moreover, the serum (Figure 5a) and liver (Figure 5b) triglyceride levels were compared between the mice, and the results indicated significantly higher levels (1.7- and 1.9-fold, respectively) among treated mice than controls. Taken together, these results showed that doxepin accelerated hepatic lipogenesis upregulation and, thus, increased HFD-induced fatty liver scores considerably.

### 2.6. Influences on Insulin Level and Glucose Tolerance

Next, we assessed how doxepin treatment affected mouse glucose homeostasis by performing an intraperitoneal (IP) glucose tolerance test (IPGTT) after 49 treatment days. The results (Figure 6a) indicated that doxepin induced significantly greater glucose tolerance impairment compared with that observed in the controls. Notably, doxepin treatment led to significant increases in fasting blood glucose levels compared with the control data. Moreover, at 30, 60, 90, and 120 min following injection with glucose, the treated mice demonstrated significant increases in fasting blood glucose levels relative to the control mice; in particular, compared with the fasting blood glucose levels before IP glucose injection, those at 120 min after injection were 1.2-fold higher.

Moreover, the area under the curve (AUC) of the IPGTT results over the 120-min test period significantly increased (by 23%) in the treated mice compared with the control mice (Figure 6b). When the glucose intolerance criterion was set as blood glucose > 10 mmol/L at 120 min after injection, most of the treated mice were found to be glucose intolerant (Figure 6c). Finally, the treatment group exhibited significantly lower serum insulin levels than did the control group (Figure 6d). Taken together, our results are indicative of the treated mice having developed doxepin-aggravated diabetes symptoms, with the worsening of hyperglycemia and hypoinsulinemia-associated impairment in glucose tolerance.

### 2.7. Effects on IS and IR and Phosphorylated Akt and Glucose Transport 4 Expression

Next, we evaluated the IS and IR in our mice by calculating their IS and homeostatic model assessment of IR (HOMA-IR) indices, respectively, and noted significantly increased HOMA-IR (Figure 7a) and decreased IS (Figure 7b) indexes in the treatment group compared with the control group: the HOMA-IR index was 2.4-fold higher, whereas the IS index was 2.1-fold lower.

To characterize the mechanisms underlying the observed glucose homeostasis after doxepin treatment, we evaluated phosphorylated Akt ( hosphor-Akt) and glucose transport 4 (GLUT4) expression levels in the skeletal muscles of our mice (Figure 7c); the results indicated significant increases in the expression of both proteins (31% and 58%, respectively) in the treated mice (Figure 7d,e, respectively).

### 2.8. Effects on Organ and Tissue Cr Levels and Urinary Cr Loss

Cr (III) is essential in glucose homeostasis, in which it potentiates insulin activity and improves IR [37,39]. Therefore, we evaluated whether doxepin induced changes in the Cr levels in our mice by measuring the Cr content of various harvested tissues (Table 2). According to the results, the Cr intake was generally 1.4-fold higher in the treated mice than in the control mice—a result corroborating the observation of hyperphagia in the treated mice. However, after doxepin treatment, the treated mice had 45%, 53%, 27%, 24%, and 23% lower Cr content in their blood, bone, liver, muscle, and WAT but 1.8- and 1.9-fold higher Cr content in their kidneys and urine, respectively, than did the control mice.

### 2.9. Effects on Renal Injury, Serum Creatinine and Blood Urea Nitrogen Levels, and Kidney Reactive Oxygen Species, and Antioxidant Enzyme Levels

Obesity, hyperlipidemia, and diabetes can result in renal injury [8,42]. Thus, we next performed H&E staining on kidney tissue sections to evaluate whether doxepin can induce renal damage; the treated mice had more glomerulonephritis with inflammatory cell infiltration than did the controls (Figure 8a). We also noted concurrent significant increases in serum creatinine and blood urea nitrogen (BUN) levels (by 1.6-fold and 2.1-fold, respectively) in the treated mice compared with the control mice (Figure 8b,c, respectively).

Reduction in the activity of renal antioxidant enzymes (e.g., catalase, glutathione peroxidase [GPx], and superoxide dismutase (SOD)) may lead to renal injury, whereas increases in this activity may decelerate renal nephropathy and enhance renal function [43]. Accordingly, the treated mice demonstrated 43%, 30%, and 36% decreased catalase (Figure 8d), GPx (Figure 8e), and SOD (Figure 8f) activity relative to the control mice, respectively.

In diabetes, renal ROS overproduction predominantly accelerates renal inflammation, negatively affects renal structure and function and, finally, results in diabetic kidney disease [44]. Here, the treatment group exhibited 1.5-fold higher renal ROS levels than those of the control group (Figure 8g).

## 3. Discussion

We investigated how doxepin affects the development of obesity in C57BL/6J mice receiving an HFD, and the results revealed that the obese mice in the 56-day doxepin treatment group exhibited higher obesity development rates, increased visceral fat levels, a larger AUC at 120 min after glucose injection, and aggravated glucose intolerance, IR, renal damage, and fatty liver development. Moreover, their daily food efficiency; fatty liver scores; liver, kidney, and WAT weights; adipocyte size; serum and liver triglyceride levels; and serum BUN and creatinine levels were increased; by contrast, their renal catalase, GPx, and SOD levels were reduced. Thus, the findings confirmed that doxepin has the ability to increase food intake and body weight gain and attenuate glucose homeostasis with hypoinsulinemia. Doxepin could be linked to liver adipogenesis as well as increased fatty liver scores based on the evaluation results for the associated proteins, including *FABP4* and *SREBP1* mRNA expression as well as adiponectin, FASN, and PNPLA3 activation. High IR is typically associated with a lower insulin signaling protein activity; in our doxepin-treated mice, the impairment in glucose homeostasis was exacerbated by reductions in phospho-Akt and GLUT4 expression. Moreover, at the end of the 56 treatment days, the Cr content was significantly lower in the bone, blood, liver, muscle, and WATs but significantly higher in the kidneys and urine of the treated mice than in those of the control mice. Moreover, the treated mice also exhibited decreased renal antioxidant enzyme levels and increased renal ROS levels and urinary Cr loss—corresponding to the development of hyperglycemia-associated renal injury.

In the mice on a 12-week HFD, the HFD contributed to obesity development in general; moreover, doxepin promoted body, RWAT, EWAT, kidney, and liver weight gain. Moreover, the body fat increases may have been attributable to decreased uncoupling protein 1 (UCP1) mRNA expression in brown adipose tissue (Appendix A), and UCP1 can increase energy expenditure by regulating the metabolic rate [45]. In addition, leptin—essential in food intake regulation in the hypothalamus—is primarily synthesized in WAT [27]. An increase in food intake typically leads to body weight gain. However, our doxepin-treated mice demonstrated a higher food intake and higher serum leptin levels. Thus, doxepin treatment may have inhibited the leptin signaling pathway; this assumption is supported by studies indicating that leptin signaling is inhibited by antipsychotic drugs such as risperidone, clozapine, and olanzapine, which modulate the expression of suppressors of cytokine signaling 3 and 6 genes through adenylate cyclase–mediated extracellular signal–regulated kinase activation [14,46,47]. Furthermore, the decrease in leptin receptor expression noted in our doxepin-treated mice is a leptin resistance marker (Appendix A). Taken together, these notable findings suggest that long-term doxepin use accelerates obesity development through the inhibition of leptin signaling.

Liver fat accumulation may lead to NAFLD, a serious consequence of the emerging epidemic of obesity [10,13]. In our histopathological examination, the doxepin-treated obese mice were noted to have increased fatty liver scores along with increased AST and ALT (liver injury markers) levels. The elevations in ALT and AST levels were linked to increased liver fatty infiltration [8,10,40]. We also analyzed the mRNA expression of *FABP4* and *SREBP1*, which is linked to liver lipid storage, liver steatosis, liver lipogenesis, and NAFLD pathogenesis [39,48]. In this study, long-term doxepin treatment was revealed to increase liver *FABP4* and *SREBP1* mRNA levels, which caused liver damage through increases in lipid infiltration in the liver and concurrently increased serum ALT and AST levels. Similarly, some studies have reported the occurrence of liver abnormalities, including aminotransferase abnormalities and cholestatic hepatitis, after tricyclic antidepressant treatment in patients [49,50]. In addition, a pharmacological report linked higher serotonin levels promote liver lipid accumulation and liver steatosis development [51]. These findings corroborate our results for doxepin-treated mice regarding the large increases in WAT and liver weights, serum and liver triglycerides levels, and fatty liver scores in response to increased serum serotonin levels (Appendix A). Thus, to ensure safety during clinical trials, when doxepin is used for the treatment of patients with depression who have an underlying liver disease, regular monitoring of serum ALT and AST levels is essential—even though no formal recommendation for such investigations had been made.

Here, doxepin treatment was noted to increase FASN and PNPLA3 expression in our obese mice. FASN and PNPLA3, involved in de novo lipogenesis and the liver’s accumulation of fat, can abolish IS [10]. Moreover, increased liver fat accumulation is the factor most associated with NAFLD development [8,13]. Compared with the saline-treated control mice, the mice receiving doxepin had increased fatty liver scores—possibly because doxepin reduced the activity of fibroblast growth factor (FGF)-21 (Appendix A), the physiological energy sensor associated with fat and glucose metabolism through the promotion of mitochondrial oxidative capacity and peroxisome proliferator–activated receptor-γ activity. FGF-21 can have implications for viable therapeutic strategies to reduce body weight and increase energy expenditure as well as stop or undo liver fibrosis, inflammation, and fat infiltration by increasing adiponectin secretion to inhibit the nuclear factor κB inflammatory signaling pathway [51,52]. Our results regarding the body weight and fatty liver score increases after doxepin treatment in obese mice are potentially attributable to the reduction in the serum levels of FGF-21, which attenuated protection against NAFLD (including nonalcoholic fatty liver and nonalcoholic steatohepatitis) [40,52]. Furthermore, our results that doxepin treatment increased serum ALT and AST levels accords with the inability of FGF-21 to reduce the increased lipid accumulation in hepatocytes, resulting in lipid-overload stress and the subsequent release of multiple proinflammatory factors such as tumor necrosis factor (TNF) α and C-reactive protein (CRP) [51,53]—all leading to chronic low-grade liver inflammatory status. In addition, PNPLA3 may promote liver inflammation during NAFLD progression by increasing TNF-α expression and activating the endoplasmic reticulum stress-mediated and NF-κB–independent inflammatory inositol–requiring enzyme-1α/c-Jun amino-terminal kinase pathway [54]. In the last step of fatty acid biosynthesis, FASN exerts catalytic activity. Thus, FASN is potentially a prominent factor determining the maximum liver capacity of fatty acid generation through liver lipid accumulation, a process which begins with simple liver steatosis and possibly progresses to inflammation [55]. Taken together, our results indicate that doxepin induced high liver lipid accumulation and that it aggravated liver damage in diet-induced NAFLD.

Tricyclic antidepressants treatment can increase glucose levels and reduce insulin levels in animals and humans, thus increasing metabolic risks in nondiabetic patients with depression or deteriorating glucose homeostasis and aggravating glucose intolerance in T2DM patients with depression [28,56,57]. Moreover, evidence regarding tricyclic antidepressants such as doxepin and amitriptyline has indicated that hyperglycemia develops from pancreatic insulin release being inhibited [58] and that tricyclic antidepressant use inhibits glucose transport, resulting in decreased glucose uptake [59]. The findings in our doxepin-treated mice are consistent with previous findings regarding aggravated glucose intolerance and decreased insulin levels. Here, long-term doxepin use reduced serum insulin levels, corroborated by the reduction in the proportion of β cells in the pancreatic islets (Appendix A). IPGTT was then used to investigate how doxepin affected hyperglycemia in our HFD-fed mice; the doxepin-treated mice were noted to have considerably exacerbated glucose intolerance, based on the increased 120-min AUC for plasma glucose levels and HOMA-IR index along with the decreased IS index. This observation is potentially attributable to the decreases in the skeletal muscle phospho-Akt expression in the doxepin-treated mice. In general, low phospho-Akt and serum FGF21 levels (Appendix A) can lead to attenuated insulin signaling, which can, in turn, impair glucose homeostasis and enhance IR [10,40]. Glucose intolerance aggravation after doxepin treatment could also be attributable to decreased muscle expression of GLUT4, the enzyme involved in the rate-limiting step in muscle glucose metabolism [11,13]. Considerable decreases in basal glucose transport are indicative of severe IR and glucose intolerance in mice with T2DM, who selectively lack GLUT4 in their muscles [10]. Taken together, our results show that our doxepin-treated mice exhibited severe hyperglycemia and suppressed IS.

Long-term doxepin use can promote hyperglycemia induced by decreases in insulin signaling. In contrast to the current results, those of some studies have indicated that doxepin treatment can attenuate hyperglycemia and increase IS and insulin secretion in humans [26,27] and animals [28,29]. However, severe hypoglycemia after short-term (14-day) doxepin use has been associated with stable glucose levels in humans [26]. In rabbits, these levels have been noted to peak from 4 to 10 h postadministration, followed by complete attenuation of initial hypoglycemia at seven days postadministration and, finally, hyperglycemia at 21 days postadministration [28]. In addition, the doxepin treatment duration in our study (eight weeks) was longer than that reported by Chen et al. [29], who found that HFD-fed mice treated with doxepin for four weeks had improved glucose intolerance in their oral GTT. These conflicting findings are potentially attributable to variance in doxepin treatment duration or some antidepressants (e.g., maprotiline, bupropion, fluoxetine, fenfluramine, imipramine, and sertraline) increasing IS through increases in insulin secretion over a short period [60]. Therefore, in this study, we included variables to reflect the hypoglycemia- or hyperglycemia-inducing biases, which may occur in patients with different exposure periods. Moreover, excessive insulin secretion along with hyperglycemia possibly results in a negative effect on pancreatic β-cell function; this is followed by β-cell expansion failure and eventually β-cell failure and diabetes [61]. Thus, T2DM patients using antidepressants should pay attention to any hypoglycemia or hyperglycemia symptoms and follow strict blood glucose self-monitoring.

Cr, an essential mineral stored in bones, is released and distributed when required for regulating tissue-specific glucose uptake in metabolic organs or tissues, including the liver, skeletal muscle, and adipose tissues; it, thus, facilitates insulin signaling activation and thus ensures homeostatic blood glucose control [37,62]. This Cr movement has a metabolic role, whereby it controls glucose metabolism. In our study, without any insulin treatment, significantly decreased Cr content in the blood, muscle, liver, bone, and WAT of the mice in the treatment group was observed compared with that of the mice in the control group, even though higher Cr intake is linked to hyperphagia after doxepin treatment. These findings confirm that doxepin treatment negatively affects Cr accumulation in the aforementioned tissues [13]. This result corroborates the significant decreases in serum Cr concentrations in patients with uncontrolled T2DM [63]. Thus, doxepin potentially played a regulatory role in our study regarding the distribution of Cr in the tissues of obese mice. Moreover, in our mice, the Cr content in the liver and muscles (the locations of highest incidence of glucose metabolism) decreased after doxepin treatment, which was concomitant with hyperglycemia occurrence [32,46]. This decrease in Cr content could have compromised the hyperglycemia alleviation capacity after doxepin treatment in our mice. Thus, long-term doxepin administration may alter Cr content differently in various organs and tissues; still, hyperglycemia exacerbation in our doxepin-treated mice was observed from Cr redistributing or mobilizing.

A considerable amount of trace elements are lost through urine, but most of these are reabsorbed at the proximal renal tubule [13]. In our treated obese mice, although the Cr content was low in the blood and metabolic tissues, it was high in the kidneys and urine. Thus, doxepin must have caused Cr to be released from metabolic tissues into the kidney, increasing the urinary loss of Cr. In general, increases and decreases in the amount of trace metals excreted through urine and reabsorbed, respectively, occur as renal damage responses [37]. Our observations were supported by the increased values of kidney function indexes (i.e., BUN and creatinine) in our treated mice. We also noted that doxepin treatment induced the formation of injury lesions in the kidneys, along with glomerulonephritis. The increase in urinary Cr loss and negative total Cr balance in the doxepin-treated mice likely occurred in association with the aforenoted injuries. Considerable urinary Cr loss led to the aggravation of glucose intolerance. Thus, determining whether related drugs other than doxepin—which may be administered in combination with or as a replacement for doxepin to reduce renal side effects—lead to the aforementioned effects in depressed patients with T2DM is essential.

A clinical study reported increased inflammation along with CRP elevation in patients with depression on tricyclic or tetracyclic antidepressants [64]. Moreover, systemic chronic inflammation can accompany obesity and hyperglycemia, and this type of inflammation has a crucial role in the T2DM development process [8,11]. Numerous clinical and experimental studies have indicated that chronic increases in serum acute-phase reactant CRP, interleukin (IL) 1β, and TNF-α levels are associated with IR [8,65,66]. Accordingly, we noted that doxepin treatment led to increases in serum CRP, IL-1β, and TNF-α levels in our doxepin-treated mice, in parallel with glucose intolerance exacerbation (Appendix A). Alternatively, hyperglycemia contributed to Kupffer cells activating and then secreting proinflammatory cytokines such as TNF-α and IL-6 [67,68]. Higher TNF-α levels lead to liver injury mediated by hepatocellular apoptosis [69]. Indeed, in our treated mice, doxepin caused hyperglycemia; we noted increased expression of activated Kupffer cells (Appendix A) during immunohistochemical (IHC) staining of liver tissues with F4/80 (Kupffer cell–specific marker) [70] along with elevations in serum ALT and AST levels in the doxepin-treated mice. These data clearly indicate that doxepin plays a key role in cytokine-mediated obesity, hyperglycemia, and liver damage.

A noted side effect of long-term tetracyclic antidepressant use in patients with cancer is chronic renal failure [71]; the reason underlying this effect is that these drugs have adverse effects, including obesity, metabolic syndrome, or diabetes, all of which are correlated with renal impairment to some degree [8,72,73]. Hyperglycemia can cause glomerular, vessel, and tubular injury in the kidneys, all of which may cause diabetic nephropathy in correlation with renal inflammation, further resulting in CKD [74,75]. Similarly, in our treated mice, doxepin caused glomerulonephritis—characterized by the presence of inflammation in renal glomeruli and elevations in renal function index values [76]—as an adverse effect; we noted increased expression of renal inflammatory cytokines (such as IL-1β) in the IHC staining of kidney tissue (Appendix A), along with elevations in serum creatinine and BUN levels in doxepin-treated mice. Thus, clinicians should be aware of the possible adverse consequences of renal damage in patients receiving doxepin therapy and practice timely renal function monitoring.

Hyperglycemia also leads to ROS generation; in turn, ROS cause lipid peroxidation and damage in diabetic rat brains and kidneys [77]. As noted, doxepin is a risk factor for glucose intolerance; thus, the doxepin-treated mice with hyperglycemia in this study showed a reduction in antioxidative defense enzyme levels in the kidneys. This phenomenon can be explained by the low levels of antioxidative enzymes being insufficient for protecting against free radical damage, which may cause renal inflammation [78]. Higher renal ROS levels and lower SOD, GPx, and catalase levels were noted in the kidneys of our doxepin-treated mice, along with renal impairment. However, studies have reported that antidepressants exert antioxidant effects by acting as ROS scavengers in silico [79] or by increasing antioxidant and detoxification enzymes’ mRNA expression in the early embryonic development stages of zebrafish [80]. The discrepancy noted between the current (in vivo, animal) and previous (in silico or aquatic animal) results most likely reflect effects of the presence of a hyperglycemic illness. Thus, patients with a depressive disorder who use doxepin in the long term have increased renal damage risks than do patients not receiving long-term doxepin treatment.

Here, we could successfully generate a mouse model of obesity as well as hyperglycemia by administering an HFD and then continually administering doxepin to this model so as to promote body weight gain and increase food intake; fatty liver scores; serum ALT and AST levels; liver, kidney, and WAT weights; serum and liver triglyceride levels; and adipocyte size. These findings are possibly corroborated by lipid accumulation and nutrient metabolism regulation–related lipogenesis activation. Furthermore, doxepin administration to our mice was noted to exacerbate hyperglycemia and accelerate the progression of glucose intolerance along with glomerulonephritis. The decreases in skeletal muscle phospho-Akt and GLUT4 expression possibly contributed to the aforementioned decrease in glucose metabolism. Moreover, doxepin was noted to change Cr distribution among the collected tissues and increase urinary Cr loss. The presence of attenuated glucose homeostasis and IS but enhanced IR was thus apparent. Furthermore, in our mice, the HFD was noted to decelerate glucose homeostasis, whereas doxepin accelerated T2DM symptom development. In summary, doxepin has potential adverse effects linked to diabetes, renal damage, and fatty liver disease, and its long-term administration possibly hinders obesity control and exacerbates diabetes and CKD.

## 4. Materials and Methods

### 4.1. Animals, Diet, and Doxepin Treatment

Male C57BL/6J mice (age, five weeks) were provided by the Education Research Resource, National Laboratory Animal Centre, Taiwan; moreover, the mice were handled according to the Guidelines for the Care and Use of Laboratory Animals, as recommended by the Taiwan government. The protocol for experiments was reviewed and approved by the Institutional Animal Care and Use Committee of National Chiayi University (protocol code 109019). An HFD (diet 592Z (composition: lab-modified with 35.5% lard, 1.12 µg/g Cr, 20.4% protein, and 4.5 kcal/g metabolizable energy); PMI Nutrition International, Brentwood, MO, USA) was continually administered to the mice for 12 weeks—a duration longer than that used previously (four weeks) for inducing obesity [10].

We then separated the mice into two groups: one treated with 5 mg/kg doxepin (Sigma-Aldrich, St Louis, MO, USA) and another treated with saline (vehicle control) via daily gavage for the final 56 days of the HFD-feeding duration (39.38 ± 0.96 g for treated mice vs. 39.48 ± 1.23 g for control mice, *p* > 0.05). Here we based the doxepin dosage on mouse studies on how doxepin affects glucose homeostasis, behavior, depression, stress, and memory deficits [29,81,82]. In our preliminary investigation, obese mice were orally administered 2 mg/kg doxepin; however, their body weight and body weight gain at this dosage did not differ significantly from those of the obese control mice (Appendix A). Consequently, we began this study with a doxepin dosage of 5 mg/kg in our mouse model.

Each mouse was kept in a microisolation cage on HEPA-filtered ventilated racks under controlled temperature (22 ± 1 °C) and humidity (55% ± 5%) conditions with a 12-h light–dark cycle and provided with ad libitum access to water and food. Their food intake and body weight were monitored each week from the beginning of the experiment onward. When the experiment was over, we anesthetized the mice and then harvested various tissues and extracted blood for subsequent analysis. Oral doxepin’s effects on body weight, food intake, blood glucose level, endocrine profile, fatty liver score, biochemical change, liver triglyceride level, insulin signaling expression, adipocyte content, and renal pathology were also evaluated. Finally, we kept all the mice in separate metabolic cages (SN-783-0; AS ONE, Osaka, Japan) for 12 h and then killed them for urine collection.

### 4.2. Measurement of Body Weight, Food Intake, and Insulin and Leptin Levels 

Body weight and food intake were measured weekly throughout the study period. For food intake measurement, the food left inside the food dispenser or spilled on the floor of the cages was weighed. Next, we measured serum insulin and leptin levels by using enzyme-linked immunosorbent assay (ELISA) kits for mouse leptin and insulin (#90030 and #INSKR020, respectively) from Crystal Chem (Downers Grove, IL, USA).

### 4.3. Serum Triglyceride, ALT, AST, Creatinine, BUN, and Liver Triglyceride Level Measurement

We measured the serum levels of triglycerides, ALT, AST, creatinine, and BUN from the collected blood samples on a Catalyst One chemistry analyzer (an automated chemistry analyzer from IDEXX Laboratories, Westbrook, ME, USA) by using commercial kits in accordance with the manufacturer’s instructions—the coefficient of variation between and within analysis runs was set as <2%.

As reported previously [8,10], we first extracted triglycerides from homogenized liver samples by using Triton X-100 and then solubilized these triglycerides through two gradual cycles of increasing the temperature to 90 °C over 5 min and then reducing the temperature to room temperature; this extracted solution subsequently underwent centrifugation for the elimination of any insoluble materials. We finally used the supernatant for colorimetry-based triglyceride analysis with a triglyceride quantification kit (BioVision, Milpitas, CA, USA).

### 4.4. IPGTT

At 49 days after doxepin or saline treatment initiation, we performed IPGTT with 1 g of glucose/kg body weight administered as an IP injection to mice who had been fasted overnight with free access to water. Next, we extracted blood at 0, 30, 60, 90, and 120 min postinjection from the tail vein and then analyzed glucose levels in these blood samples on a One Touch glucose meter (LifeScan, Milpitas, CA, USA). Finally, we calculated the glucose tolerance based on the AUC over 0–120 min.

### 4.5. Morphometric and Histological Tissue Analyses

We measured the absolute weights of the heart, liver, spleen, kidney, RWAT, and EWAT (expressed as percentages of the TBW).

We visualized fat infiltration in H&E-stained liver slices and then scored it as 0, 1, 2, 3, and 4 when no, <5%, 5–25%, 25–50%, and >50% liver fat infiltration was visible, respectively [11,13,38]. Furthermore, H&E-stained RWAT and EWAT sections were systematically analyzed to determine adipocyte sizes. For each section, we analyzed at least 10 fields (representing approximately 100 adipocytes) [10,11,13]. Next, the kidneys were bisected along the longitudinal axis of each kidney and then stained with H&E. We evaluated the occurrence of glomerulonephritis by blinding to the origin of the specimens, as described previously [8].

All images were acquired under Moticam 2300 (a high-resolution digital microscope from Motic Instruments, Richmond, BC, Canada). For analysis of the adipocyte size distribution, we used Motic Images Plus 2.0.

### 4.6. IR and IS Indexes

To evaluate post–doxepin treatment IR and insulin function in our mice, we used HOMA-IR and IS indexes; these indices are based on fasting blood glucose according to the HOMA method, which has been validated against clamp measurements [8,9,10,11]:HOMA-IR = [fasting insulin (mU/L) × fasting glucose (mmol/L)]/22.5(1)
IS index = [1/fasting insulin (mU/L) × fasting glucose (mmol/L)] × 1000(2)

### 4.7. RNA Extraction and Real-Time Quantitative Polymerase Chain Reaction

We extracted total RNA from liver samples by using TRI Reagent from Sigma-Aldrich. The concentration of the extracted RNA was then determined on the basis of absorbance values at 230–260 and 260/280 nm on a Qubit fluorometer from Invitrogen (Carlsbad, CA, USA). Next, we reverse transcribed 1 μg of *F**ABP4* and *SREBP1* mRNA to the corresponding cDNA by using an iScript cDNA Synthesis Kit from Bio-Rad (Hercules, CA, USA) according to the manufacturer’s instructions.

Finally, a real-time polymerase chain reaction (PCR) of the reverse-transcribed cDNA was performed with Bio-Rad’s iTaq Universal SYBR Green Supermix in accordance with the manufacturer’s instructions. We then detected the expression levels on a Bio-Rad’s CFX Connect Real-Time PCR System and calculated them relative to *ACTB* levels by using the 2^−ΔΔCt^ method. During the PCR, the thermocycle reactions were as follows: 95 °C for 5 min and then 45 cycles of 95 °C for 15 s and 60 °C for 25 s. The forward and reverse primer sequences were, respectively, 5′-GATGAAATCACCGCAGACGACA-3′ and 5′-ATTGTGGTCGACTTTCCATCCC-3′ for *FABP4* [83] and 5′-CGG AAGCTGTCGGGGTAG-3′ and 5′-GTTGTTGATGAGCTGGAGCA-3′ for *SREBP1* [8].

### 4.8. Western Blot Analysis

When the experiment was over, all mice were euthanized with anesthetic overdose in combination with CO_2_. Next, their liver and gastrocnemius muscles were quickly removed, minced coarsely, and homogenized.

We subsequently performed Western blot analysis, as described previously [13]. Here, anti-FASN, anti-phospho-Akt (Ser473), anti-GLUT4, anti-actin, and anti-Akt antibodies from Cell Signaling Technology (Beverly, MA, USA) and antiadiponectin and anti-PNPLA3 antibodies from Sigma-Aldrich were used. To enhance and detect immunoreactive signals, we employed enhanced chemiluminescence reagents from Thermo Scientific (Rockford, MA, USA) and UVP ChemStudio from (Analytik Jena, Upland, CA, USA), respectively. Finally, quantification of phosphorylation and protein expression was performed with the software program ImageJ from the National Institutes of Health (Bethesda, MA, USA).

### 4.9. Analysis of Cr Concentration

Blood, urine, and tissue (bone, liver, muscle, fat, and kidney) samples were collected from all mice at the end of the experiment. Next, all collected tissues samples were rinsed with saline, blotted dry, and weighed.

We then determined Cr levels in the collected samples as described previously [34]. In brief, these samples (25 µL of blood or urine or 0.1 g of tissues) underwent digestion at 100 °C overnight in 1 mL of 65% nitric acid. We diluted the resulting solutions by using ≤5 mL of distilled water prior to further measurement. The Cr concentrations in the resulting solution were determined through graphite furnace atomic absorption spectrophotometry on a Z-2000 series polarized Zeeman atomic absorption spectrophotometer (Hitachi, Tokyo, Japan), with 359.3 as the analysis line for Cr. Here, for each sample, the Cr concentrations are expressed in nanograms per gram and nanograms per milliliter to indicate the amount of Cr in relation to the tissue dry weight and blood or urine volume, respectively.

### 4.10. Measurement of Renal GPx, Catalase, SOD, and ROS Levels

Next, we measured the SOD, GPx, and catalase levels in fresh cortical tissue in kidney lysates obtained from homogenization of the extracted kidneys in 0.1 M Tris/HCl (pH 7.4; containing 5 mM β-mercaptoethanol, 0.1 mg/mL phenylmethanesulfonylfluoride, and 0.5% Triton X-100), as described previously [84]. We then centrifuged the mixture at 14,000× *g* for 5 min at 4 °C and employed the resulting supernatant for measurement of the GPx, SOD, and catalase levels with commercially available colorimetric kits (#K773-100, #K762-100, and #K335-100, respectively) from BioVision (Milpitas, CA, USA) according to the manufacturer instructions.

In addition, before evaluating ROS levels, we weighed kidney tissues, homogenized them using phosphate-buffered saline on ice, and then centrifuged them at 5000× *g* for 10 min at 4 °C [85]. The resulting supernatant was used to measure ROS levels with a commercially available colorimetrical kit (#KTE71621) from Abbkine (Redlands, CA, USA) according to the manufacturer’s directions.

### 4.11. Statistical Analysis

All results in this study denote the mean ± standard deviation. We conducted *t* tests for comparisons between two groups and ANOVA followed by the post hoc Bonferroni test for comparisons among ≥2 groups. *p* < 0.05 was considered to indicate significance. We used Fisher’s exact test to identify significant differences in contingency data.

## 5. Conclusions

Long-term continual administration of doxepin in mice fed an HFD accelerated obesity development and aggravated hyperglycemia, glucose intolerance, Cr distribution changes, and renal impairment, all in parallel with decelerated metabolic homeostasis, increased fatty liver scores and Cr loss, and decreased skeletal muscle GLUT4 expression. Doxepin also caused urinary Cr loss, leading to an imbalance in Cr blood levels, which is essential in abrogating hyperglycemia. Our results are also indicative of an association between doxepin-aggravated hyperglycemia—also related to decreased phospho-Akt and GLUT4 expression in insulin signaling—and impairment in glucose tolerance, a decrease in IS, and an increase in IR; thus, decreases in renal antioxidant enzyme levels and increases in renal ROS levels may have exacerbated nephropathy. Although it has antidepressive effects, doxepin may potentiate several related metabolic abnormalities. Thus, regular monitoring of blood glucose and liver and kidney functioning is recommended after the initiation of doxepin treatment. In particular, these symptoms may appear early in schizophrenic patients with cachexia or obesity, hyperglycemia, and CKD.

## Figures and Tables

**Figure 1 pharmaceuticals-14-00267-f001:**
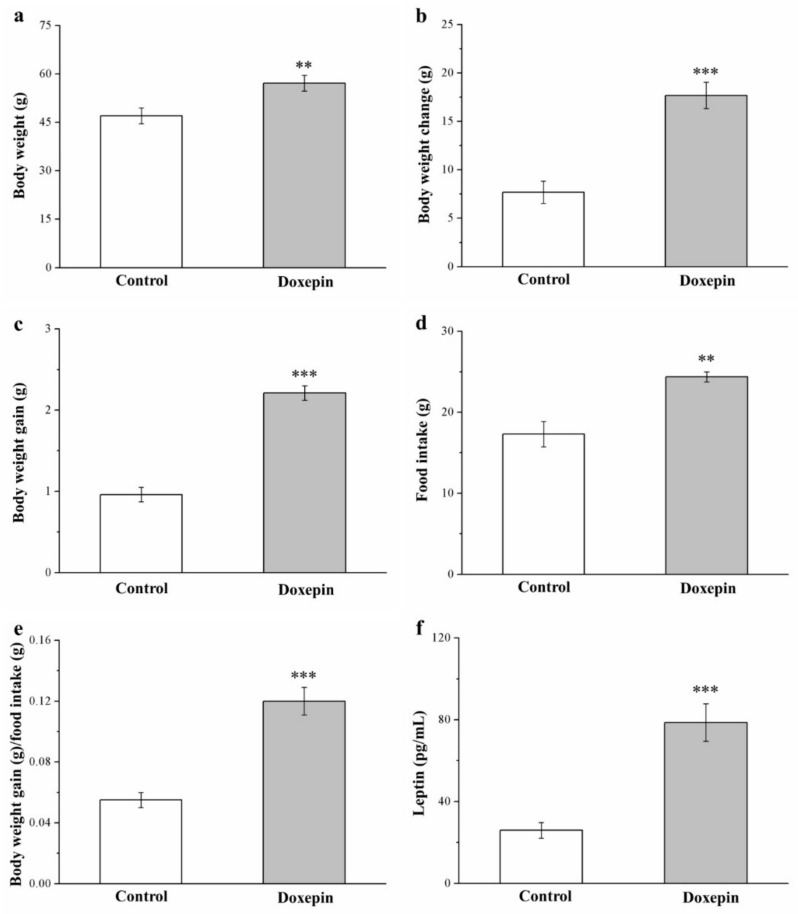
(**a**) Body weight, (**b**) body weight change, (**c**) body weight gain, (**d**) weekly food intake, (**e**) daily food efficiency, and (**f**) serum leptin levels in control and doxepin-treated obese mice over 56 days of treatment. All values represent means ± standard deviations (*n* = 10 for all groups). ** *p* < 0.01 and *** *p* < 0.001 indicate high and very high statistical significance, respectively.

**Figure 2 pharmaceuticals-14-00267-f002:**
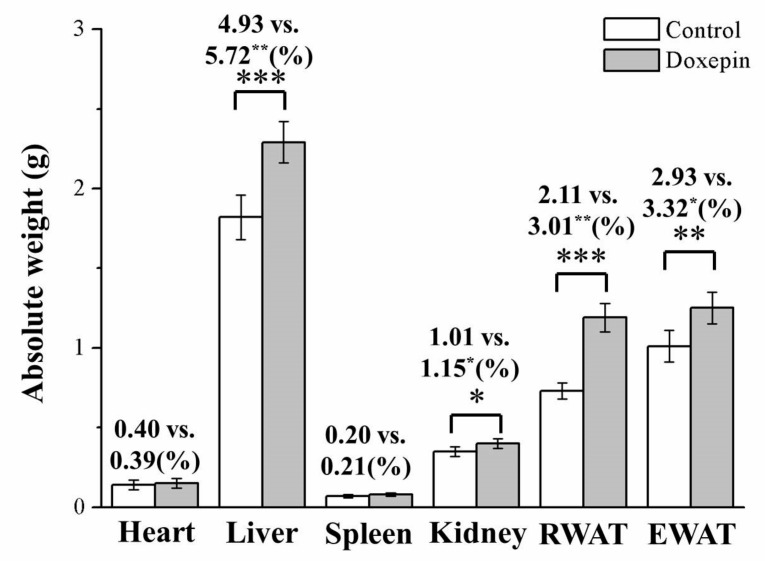
Absolute weights (i.e., percentages of the TBW) of the heart, spleen, liver, kidney, EWAT, and RWAT in control and doxepin-treated obese mice over 56 treatment days. All values represent means ± standard deviations (*n* = 10 for all groups). * *p* < 0.05, ** *p* < 0.01, and *** *p* < 0.001 indicate statistical significance, high statistical significance, and very high statistical significance, respectively.

**Figure 3 pharmaceuticals-14-00267-f003:**
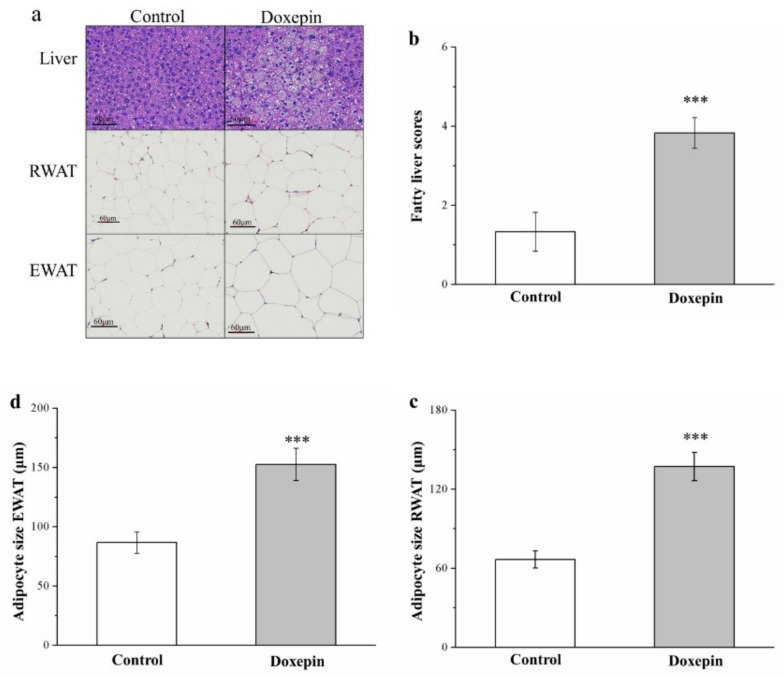
(**a**) H&E staining of livers, RWAT, and EWAT (magnification, 200×) and changes in (**b**) fatty liver score and the sizes of (**c**) RWAT and (**d**) EWAT adipocytes in control and doxepin-treated obese mice over 56 treatment days. All values denote the mean ± standard deviation (*n* = 10 for all groups). *** *p* < 0.001 indicate very high statistical significance.

**Figure 4 pharmaceuticals-14-00267-f004:**
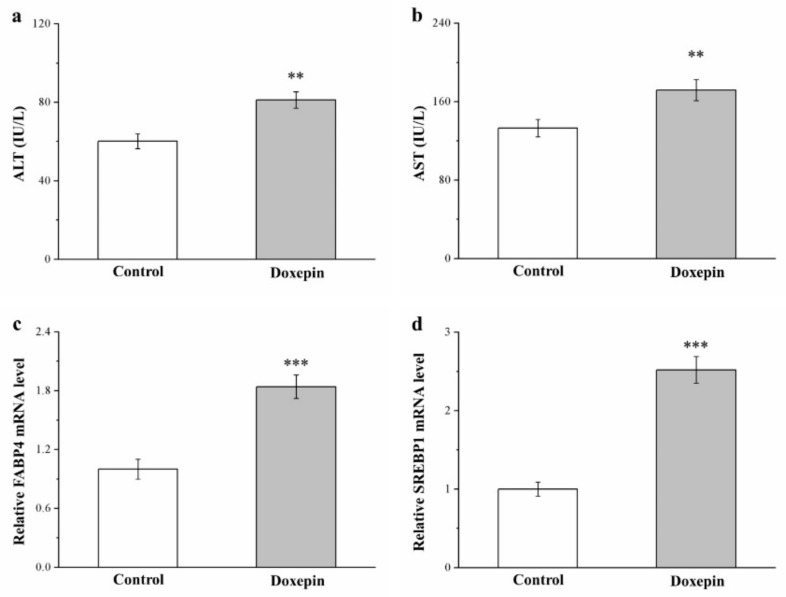
Serum (**a**) ALT and (**b**) AST levels as well as liver (**c**) *FABP4* and (**d**) *SREBP1* mRNA levels in the control and doxepin treatment groups over 56 treatment days. All values represent means ± standard deviations (*n* = 10 for all groups). ** *p* < 0.01 and *** *p* < 0.001 indicate high and very high statistical significance, respectively.

**Figure 5 pharmaceuticals-14-00267-f005:**
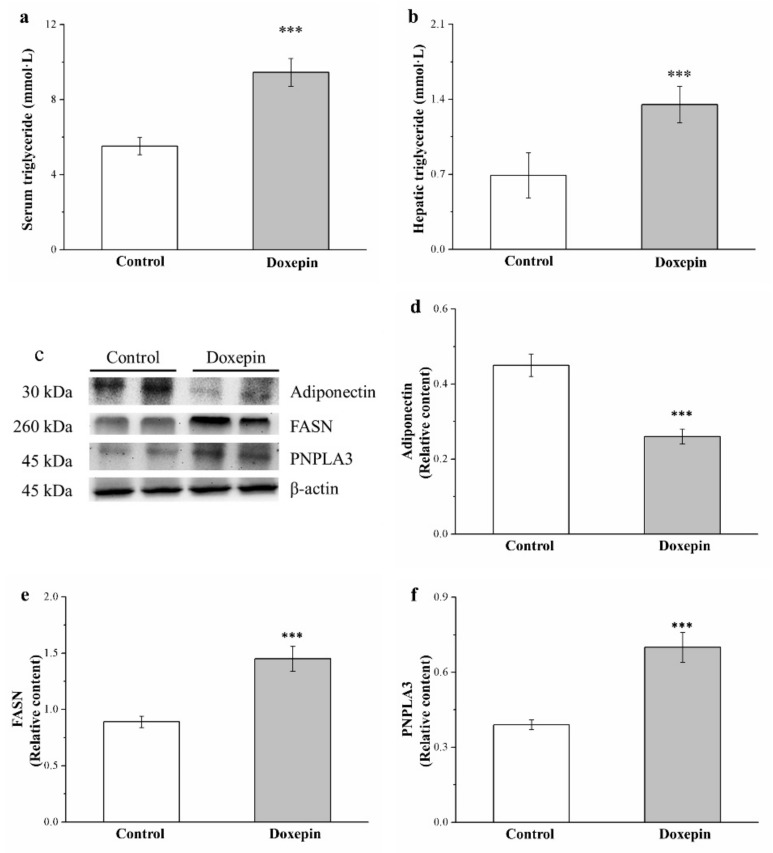
(**a**) Serum and (**b**) liver triglyceride levels, (**c**) representative Western blot of liver extracts, and (**d**) adiponectin, (**e**) FASN, and (**f**) PNPLA3 expression levels in the controls and doxepin-treated obese mice over 56 treatment days. All values represent the mean ± standard deviation (*n* = 10 for all groups). *** *p* < 0.001 indicates very high statistical significance.

**Figure 6 pharmaceuticals-14-00267-f006:**
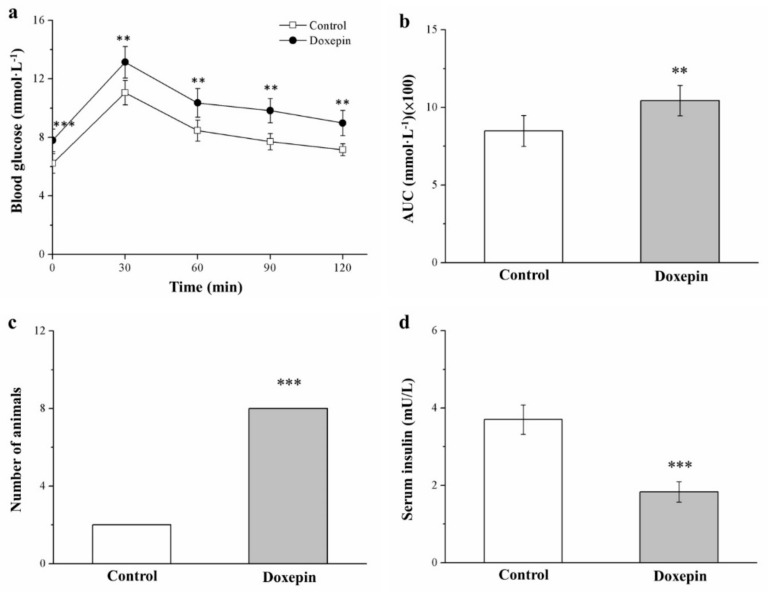
(**a**) IPGTT curve. (**b**) AUC during the 120 min following glucose injection, (**c**) glucose intolerance criterion (Fisher’s exact test), and (**d**) serum insulin levels in the controls and doxepin-treated mice over 56 treatment days. All values represent means ± standard deviations (*n* = 10 for all groups). ** *p* < 0.01 and *** *p* < 0.001 indicate high and very high statistical significance, respectively.

**Figure 7 pharmaceuticals-14-00267-f007:**
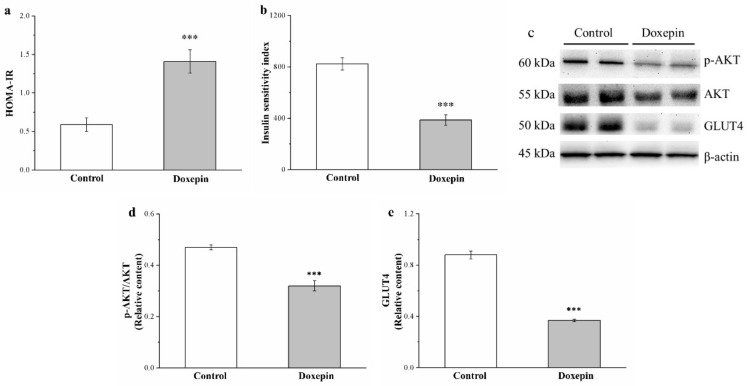
(**a**) HOMA-IR and (**b**) IS indexes, (**c**) representative Western blot of gastrocnemius muscle extracts, (**d**) phospho-Akt, and (**e**) GLUT4 expression levels of the control and doxepin-treated obese mice over 56 treatment days. All values represent the mean ± standard deviation (*n* = 10 for all groups). *** *p* < 0.001 indicates very high statistical significance.

**Figure 8 pharmaceuticals-14-00267-f008:**
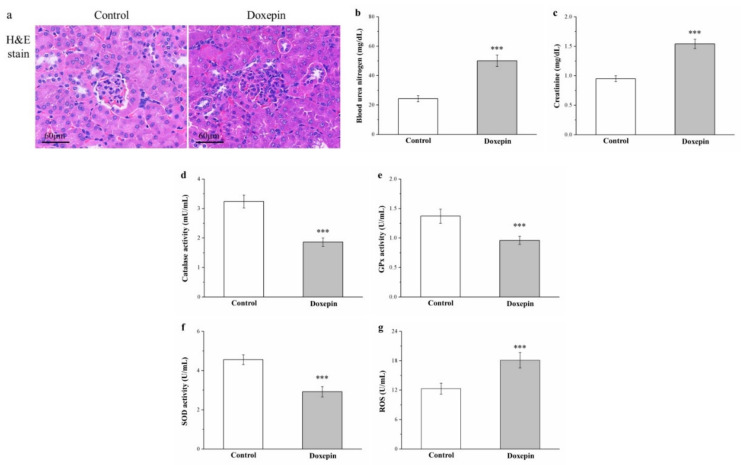
(**a**) Renal morphology (H&E staining; magnification, 200×), (**b**) serum BUN and (**c**) serum creatinine levels, (**d**) renal catalase (**e**) renal GPx, and (**f**) renal SOD activity, and (**g**) renal ROS levels in the control and doxepin-treated obese mice over 56 days of treatment. All values represent means ± standard deviations (*n* = 10 for all groups). *** *p* < 0.001 indicates very high statistical significance.

**Table 1 pharmaceuticals-14-00267-t001:** Adipocyte size distribution in control and doxepin-treated obese mice over 56 treatment days.

Variable	Control	Doxepin
RWAT		
Adipocyte diameter		
0–50 μm (%)	20.00 ± 0.65	0 ± 0 ***
50–100 μm (%)	76.36 ± 1.25	10.91 ± 1.07 ***
100–150 μm (%)	3.64 ± 0.42	50.91 ± 1.12 ***
>150 μm (%)	0 ± 0	38.18 ± 0.96 ***
EWAT		
Adipocyte diameter		
0–50 μm (%)	5.45 ± 0.47	0 ± 0 ***
50–100 μm (%)	69.1 ± 0.94	12.73 ± 0.57 ***
100–150 μm (%)	25.45 ± 1.03	36.36 ± 1.69 ***
>150 μm (%)	0 ± 0	50.91 ± 1.42 ***

All values denote the mean ± standard deviation (*n* = 10 for all groups). *** *p* < 0.001 indicates very high statistical significance.

**Table 2 pharmaceuticals-14-00267-t002:** Tissue and organ Cr levels in control and doxepin-treated obese mice over 56 treatment days.

Variable	Control	Doxepin
Chromium intake/mouse/week (μg)	19.37 ± 0.51	27.28 ± 0.82 **
Blood (ng/mL)	162.16 ± 7.67	90.43 ± 6.74 ***
Bone (ng/g)	367.19 ± 10.42	172.13 ± 8.41 ***
Liver (ng/g)	70.24 ± 6.85	51.39 ± 3.82 **
Muscle (ng/g)	50.69 ± 4.72	38.54 ± 3.25 **
Epididymal fat pads (ng/g)	45.31 ± 2.08	34.65 ± 1.94 **
Kidney (ng/g)	99.78 ± 2.81	181.63 ± 3.34 ***
Urine (ng/mL)	56.73± 2.03	105.52 ± 2.55 ***

All values represent the mean ± standard deviation (*n* = 10 for all groups). ** *p* < 0.01 and *** *p* < 0.001 indicate high and very high statistical significance, respectively.

## Data Availability

The data presented in this study are available on request from the corresponding author.

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
