# Peer review of "Doxepin Exacerbates Renal Damage, Glucose Intolerance, Nonalcoholic Fatty Liver Disease, and Urinary Chromium Loss in Obese Mice"

_pharmaceuticals, 2021, doi:10.3390/ph14030267_

Round 1
Reviewer 1 Report
This clear and well-organized document makes a significant contribution to the discipline
The authors adequately explain the results obtained and give a reasonable interpretation of the data
In my opinion this article is a good fit for this journal and I would accept it without revisions
Reviewer 2 Report
The research article “Doxepin Exacerbates Renal Damage, Glucose Intolerance, Non- alcoholic Fatty Liver Disease, and Urinary Chromium Loss in
Obese Mice” is dedicated to the analysis of doxepin affects lipid change, glucose homeostasis, chromium (Cr) distribution, renal impairment, liver damage, and fatty liver scores in C57BL6/J mice subjected to a high-fat diet and 5 mg/kg/day doxepin treatment for 8 weeks.
The article is well written.
The study has a good design.
The article is logically divided into sections and subsections.
In the article there are no grammatical and stylistic errors.
There is a large number of tables and figures of good quality presented in the article.
The methods are presented in sufficient detail. Researchers who read the article will be able to reproduce them.
The references cited relevant and adequate.
The work has a high degree of novelty.
In my opinion, this review paper can be recommended for publication after minor revision.
It is recommended to describe the methods in more detail in the sections “Western Blot Analysis”.and “Statistical Analysis”.
It is recommended to include a list of abbreviations, used in the article.
It is recommended to add articles of 2020-2021 to the list of references.

Reviewer 3 Report
In this paper, Chang and collaborators analyzed the effect of the antidepressant drug doxepin on renal damage, chromium loss, glucose intolerance and liver steatosis. The study is well-organized and the conclusions are supported by the results. However, I have a number of concerns.
- the authors stated "obese mice were orally administered 2 mg/kg doxepin; however, the differences in body weight and body weight gain were nonsignificant (Figure S9). Therefore, we adopted a doxepin dosage of 5 mg/kg thereafter." Is the 5mg/kg dosage translationally relevant? Did the authors choose this dosage based on previous publications?
- Other liver staining specific for fatty acids (e.g. Bodipy or Sirius Red) will increase the value of the work.
- Information about the involvement of hepatic stellate and Kupffer cells would be interesting
- the discussion is extremely long, and even too detailed. I would recommend to shorten the discussion and the conclusion.
